# Comparison of Myocardial Function in Young and Old Mice During Acute Myocardial Infarction: A Cardiac Magnetic Resonance Study

**DOI:** 10.3390/diagnostics15121447

**Published:** 2025-06-06

**Authors:** Antonia Dalmer, Paul Wörner, Mathias Manzke, Ralf Gäbel, Tobias Lindner, Felix G. Meinel, Marc-André Weber, Robert David, Cajetan I. Lang

**Affiliations:** 1Institute of Diagnostic and Interventional Radiology, Paediatric Radiology and Neuroradiology, University Medical Centre Rostock, 18057 Rostock, Germany; mathias.manzke@med.uni-rostock.de (M.M.); tobias.lindner@med.uni-rostock.de (T.L.); felix.meinel@med.uni-rostock.de (F.G.M.); marc-andre.weber@med.uni-rostock.de (M.-A.W.); 2Department of Cardiac Surgery, University Medical Centre Rostock, 18057 Rostock, Germany; paul_woerner@web.de (P.W.); ralf.gaebel@med.uni-rostock.de (R.G.); robert.david@med.uni-rostock.de (R.D.); 3Core Facility Small Animal Imaging, University Medical Centre Rostock, 18057 Rostock, Germany; 4Department of Life, Light and Matter, University of Rostock, 18059 Rostock, Germany; 5Department of Cardiology, University Medical Centre Rostock, 18057 Rostock, Germany

**Keywords:** cardiac magnetic resonance imaging, myocardial infarction, strain, mice, age

## Abstract

**Background/Objectives**: This study aimed to compare changes in functional and strain parameters in young and old mice using cardiac MRI before and shortly after myocardial infarction. **Methods**: In this prospective experimental study, 7 young mice and 10 old mice underwent a cardiac MRI 5 days before and 2 days after myocardial infarction by LAD ligation. Functional parameters such as EDV, ESV, EF, SV, and Strain were determined. **Results**: EDV in the young mice before LAD ligation was significantly lower than in the old mice (*p*-value 0.002). EDV significantly increased after infarction in both groups. ESV was significantly lower in young mice before infarction than in old mice (9.7 ± 2.6 vs. 13.8 ± 3.9 [µL], *p* = 0.029). After infarction, the mean value was still lower but no longer significant. There was no significant difference between young and old mice either before or after infarction for the EF. But again, the decrease was significant for both groups (old: *p* < 0.0001 and young: *p* = 0.0009). Each global strain showed deterioration after infarction. This difference was significant in both subgroups for young mice and old mice for each strain. There were no differences either before or after infarction between the young and old mice. **Conclusions**: There were differences in functional parameters between young and old mice in EDV, SV, and CO. Changes in strain parameters in the acute phase post-myocardial infarction did not differ significantly between young and old mice, while there was a clear deterioration in strain parameters after infarction in both groups.

## 1. Introduction

Cardiovascular diseases, especially acute myocardial infarction (MI), are the leading causes of death globally. Data from the World Health Organization (WHO) show that 16% of global deaths are caused by ischemic heart disease [1,2]. Age is an important risk factor for the occurrence of myocardial infarction [3,4].

Interestingly, there is usually little difference in outcomes between young and older patients [5]. These are surprising results since the prevalence of heart failure increases with age and there are changes in cardiovascular physiology. There is a thickening of the left ventricular wall with increasing age. The afterload increases. The heart rate increases to keep the cardiac output constant. Contractility also changes. Cardioprotective mechanisms and repair mechanisms become weaker [6].

Diagnosis and therapy in cardiovascular diseases have changed significantly in recent years. In addition to modern biomarkers, radiological imaging is increasingly used. Cardiac MRI (CMR), along with echocardiography, is an appropriate tool for examining cardiac function. In addition to volumetry, numerous functional parameters can be determined. Strain measurements can provide critical insights into the myocardial motion and contractility of the heart. The recorded parameters are valuable indicators for optimizing therapy and long-term outcomes in patients [7].

The mouse as an animal model is widely used due to its very similar anatomy to human hearts [8]. However, there are also challenges, especially in cardiac imaging. The mouse heart is very small, and mice have heart rates up to 10 times higher [9,10].

There is a lack of data concerning the acute phase of myocardial infarction in young vs. old individuals in both humans and mice. Therefore, the purpose of our experimental study was to compare CMRs before and shortly after myocardial infarction in young and old mice to investigate the impact of their age on functional parameters and strain.

## 2. Materials and Methods

### 2.1. Study Design and Animal Model

In this prospective experimental study, we compared young (12-week-old) mice and old (21-week-old) mice. All examined mice were females of the strain C57BL/6, purchased from Janvier Labs (Le Genest-Saint-Isle, France). The animals were kept under a 12 h/12 h day/night cycle with free access to water and food and cared for according to regulations. All the experimental protocols were approved by the federal animal care committee of LALLF Mecklenburg-Vorpommern (Germany, approval number 7221.3-1-018/21, date 5 January 2022). All experiments were performed in accordance with the relevant guidelines and regulations of the animal care committee. Efforts were made to minimize suffering, including the use of appropriate anesthesia and analgesia during surgical interventions. Animals’ well-being was monitored daily.

### 2.2. Left Anterior Descending Artery (LAD) Ligation

LAD ligation was performed by an experienced researcher with experience in small-animal MI operations. Mice were anesthetized by the intraperitoneal administration of pentobarbital sodium (50 mg/kg body weight), and the subcutaneous injection of fentanyl (0.2 mg/kg). They were intubated using a 16-gauge peripheral venous catheter (B.Braun) and ventilated using a small-animal ventilator (MiniVent Type 845, Hugo Sachs Elektronik, March, Germany). During the first three days after surgery, 16 mg metamizole per day was applied via s.c. injections, followed by the administration of the same amount via drinking water starting at day 4.

A thoracotomy and subsequent ligation were performed using a surgical microscope (OPM 241 F, Carl Zeiss, Oberkochen, Germany). The LAD was ligated using a 8-0 prolene suture. A loss of color in the supply area confirmed the success of the ligation. The thoracotomy wound was sutured with DS19 Safil^®^ (B.Braun, Melsungen, Germany).

### 2.3. Cardiac Magnetic Resonance (CMR) Imaging

MRI scans were performed 5 days before and 2 days after the LAD ligation. For MRI imaging, the mice were anesthetized with 1.5–2.5% isoflurane in oxygen. Respiration and body temperature were monitored using a physiological monitoring system (SA Instruments, Stony Brook, NY, USA). Examinations were performed with a 7 Tesla small-animal MRI system (BioSpec 70/30, maximum gradient strength 440 mT/m, 112/86 mm volume resonator in transmit mode and 2 by 2 surface array coil in receive mode, Bruker BioSpin GmbH, Ettlingen, Germany). Mice were placed in the supine position.

The MRI sequences:Planning scans;1 short-axis stack, comprising 7 independent slices from the heart basis to the apex; integrated Cine-FLASH sequence with navigator echo (Bruker); TE/TR, 2.2/8.6 ms; FA, 15°; FoV, 29.5 × 25.1 mm; matrix, 210 × 180; resolution, 140 × 140 µm; slice thickness, 1 mm; oversampling, 140; 14 movie frames; navigator parallel to slice; TA per slice, 2:46 min:s;3 long-axis cine scans: 1-, 2-, and 3-camber view (same scan parameters as short-axis stacks).

The MRI data analysis was performed with the software civ42^®^ Version 5.14.2 (2976) (Circle, cardiovascular imaging, Calgary, AB, Canada). For analysis, the borders of the epicardium were outlined using a semi-automated tool and corrected by two experienced researchers. We determined the end-diastolic volume (EDV), end-systolic volume (ESV), stroke volume (SV), and ejection fraction (EF) as functional parameters. In addition, we calculated the cardiac output (CO). As part of the strain analysis, we recorded the radial and circumferential strain on the short axis (SAX GRS and SAX GCS) and radial and longitudinal strain on the long axis (LAX GRS and LAX GLS). In Figure 1, there is a visual example of the strain evaluation. We used a global strain referring to the percentage of myocardial deformation along the axis of the left ventricle during the cardiac cycle. The strain rate describes the speed at which this deformation occurs over time. Both parameters were derived automatically using the feature-tracking module for analysis. The longitudinal strain reflects myocardial shortening from base to apex, the circumferential strain describes deformation around the short-axis plane, and the radial strain indicates myocardial wall thickening during systole.

### 2.4. Statistical Analysis

For statistical analyses, we used IBM SPSS Statistics 29.0.0.0 and GraphPad Prism 9. Calculated values for EDV, ESV, SV, EF, strain, and age are presented as mean ± standard deviation. The *t*-test was used for comparison between subgroups (young vs. old mice). The paired *t*-test was used for the comparison of values before and after myocardial infarction in the same individuals. The unpaired t-test was used for comparison between young and old mice. *p*-values <0.05 were considered significant.

We conducted an observational study aimed at providing a comprehensive assessment of myocardial structure and function. Given the descriptive nature of our approach, we did not specify a primary outcome measure.

## 3. Results

### 3.1. Functional Parameters

We examined 7 young mice and 10 old mice. The young mice were on average 83.3 ± 2.9 days old, and the old ones, 634.8 ± 49.0 days. As shown in Table 1 and Figure 2a, the EDV in the young mice before LAD ligation was significantly lower at 30.8 ± 4.4 µL than in the old mice at 43.5 ± 8.3 µL (*p*-value 0.002, Table 2). Also, the EDV significantly increased after MI in both groups. The absolute increase in EDV was 13.8 ± 8.4 µL for young mice and 10.0 ± 10.5 µL for old mice.

As shown in Table 1, the ESV was significantly lower in young mice before infarction than in old mice (9.7 ± 2.6 vs. 13.8 ± 3.9 [µL], *p* = 0.029). After infarction, the mean value was lower but without significance (*p*-value 0.418, Figure 2b and Table 2). The end-diastolic volume more than doubled in both groups; the increase was significant, as well. The absolute change in ESV was similar between age groups (delta in old mice, 19.0 ± 10.3 µL, *p*-value 0.0002, vs. young mice, 19.7 ± 9.1 µL, *p*-value 0.0012 (Table 1).

There was no significant difference between young and old mice either before or after infarction for the EF. But again, the decrease was significant for both groups (old: *p* < 0.0001 and young: *p* = 0.0009, Table 2, Figure 2c).

SV was significantly higher for old mice before infarction (Figure 2d, Table 2). Again, in the subgroups, there was a significant difference before and after infarction.

### 3.2. Global Strain

Each strain showed deterioration after infarction (Figure 3a–d). This difference was significant in both subgroups for young mice and old mice at each strain. There were no differences either before or after infarction between the young and old mice. Even so, the strain worsened for all directions and axes (SAX GRS delta −17.6 ± 6.8%, SAX GCS delta 8.1 ± 3.1%, LAX GRS delta −15.0 ± 6.2% and LAX GLS delta 8.2 ± 3.5%, Table 1).

### 3.3. Segmental Strain

Regarding segmental strain, no significant differences were observed between young and old individuals. However, a clear difference before and after infarction can be observed in both groups. This is illustrated in Figure 4a–d, where a boxplot representation shows a distinct divergence in strain between day -5 and day 2, particularly in segments 9–16. These segments correspond to the basal and mid inferior, inferolateral, and inferoseptal regions—territories typically affected in the setting of an LAD occlusion. This observation is made for all four strains. Figure 5 further supports this observation with color-coded polar maps, showing the percentage reduction in strain. Here, the greatest strain reduction is again seen in the inferior, inferolateral, and inferoseptal segments, particularly in the apical and mid-level regions. Only longitudinal strain has a slightly different pattern.

## 4. Discussion

The purpose of this study was to investigate potential differences in the acute response to myocardial infarction in young and old mice. We demonstrated that the functional parameters and strain changed in the entire collective after LAD ligation. Thereby, relevant differences between the age groups regarding the extent of these changes were only observed for some functional parameters and not for strain. However, the absence of significant differences in strain parameters between age groups may in part be due to limited statistical power, and this should be considered when interpreting the findings.

Prognosis after myocardial infarction depends on LV function. This can be determined precisely and reproducibly by MRI, as several authors have already shown [11,12,13,14].

Both young and old mice showed the deterioration of functional parameters. Ischemia causes acute cell death and dilates the ventricle, thereby increasing ESV and EDV. The SV and EF decrease. However, young mice have a superior cardiac reserve, as shown by Strait et al. [4]. Interestingly, in our study, functional parameters deteriorated very similarly. Our data suggest that young mice have a better initial situation, but this does not save them from significant damage.

Our findings have potential translational implications, particularly with regard to early therapeutic intervention following myocardial infarction. The observation that both young and old mice exhibited a comparable acute deterioration in functional parameters suggests that the early phase of ischemic injury may be similarly critical across age groups. This supports the concept that therapeutic strategies targeting the immediate post-infarction period could be effective regardless of age. Furthermore, the lack of major differences in strain behavior between young and old mice challenges the common assumption that young animal models are inherently less representative of aged human physiology. While the long-term healing and remodeling processes may still differ, our data suggest that young mouse models may provide insights into the acute functional dynamics of myocardial infarction and could be useful for the preclinical evaluation of early cardioprotective interventions [8,15].

Various studies recommend against transferring data from young laboratory animals to old patients due to differences in post-MI healing and adverse cardiac remodeling [16,17]. However, our findings do not support this point of view. LV remodeling in mice showed a trend towards even more extensive dilation in young mice.

Interestingly, we observed no significant differences in strain parameters between young and old mice either before or after myocardial infarction. This contrasts with human studies, which report age-related reductions in myocardial strain. Andre et al. have reported that radial strain differs by age and gender in humans [18]. One possible explanation is that mice, unlike humans, exhibit a limited age-related structural and functional decline in the early stages of aging, partly due to their shorter lifespan and different patterns of cardiac remodeling. Additionally, strain measurements in mice may be more influenced by technical limitations, such as a lower spatial resolution relative to wall thickness, high heart rates, and differences in myocardial fiber architecture. These factors may mask subtle age-related changes that are more readily detected in human hearts [3,19]. Indeed, Hammouda et al. have shown that strain can distinguish between healthy and diseased hearts in mice [20].

Yang et al. showed a similar outcome between two age groups [5]. To confirm this in our model, we will next study the mice over a longer time course.

Due to the limited number of variables and the exploratory nature of the study, no ANOVA or correction for multiple comparisons was applied. This should be considered when interpreting the results.

### Limitations

We examined a small sample size, which, however, is not uncommon in mouse studies. The inference to humans is limited due to differences in cardiac anatomy and physiology. However, the mouse model is well established and accepted. Epidemiological studies for comparison will be useful in the future. The use of feature-tracking strain analysis in mice with high heart rates (~500 bpm) is technically complicated. Although imaging parameters were optimized and the heart rates derived from cine loops matched externally measured values, the high frequency may still limit the temporal resolution and accuracy of strain assessment. Nevertheless, similar approaches have been successfully applied in previous murine studies, demonstrating the general feasibility of CMR-based strain analysis in small-animal models [21].

The software used for strain analysis (Circle CVI, civ42® Version 5.14.2 (2976)) is not formally validated for murine models. Although the anatomical settings were adapted in consultation with the manufacturer to match murine cardiac geometry, species-specific validation data are lacking. Therefore, the strain results should be interpreted with appropriate caution. The software Circle for CMR analysis had already been used by other research groups in mice [22]. However, plausible results could be achieved by meticulously checking and, if necessary, correcting all contours. As an additional limitation, we did not perform gadolinium-enhanced scar quantification. However, our primary focus was on the acute functional response rather than on tissue characterization. This study focused on the acute effects of infarction. Further studies are warranted to address potential differences in the remodeling response of young vs. old mice over a longer follow-up.

A limitation of this study is the exclusive use of female mice, which may restrict the generalizability of the findings and does not allow it to account for potential sex-related differences in cardiac function and post-infarction remodeling.

Furthermore, no reperfusion model was used in this study. As a result, only the consequences of ischemia could be investigated. This approach does not allow for conclusions regarding potential age-related differences in the response to reperfusion injury. However, previous studies have demonstrated age-related differences in various reperfusion models, highlighting the importance of further investigation in this context [16,23].

A limitation of the present study is the absence of histological confirmation of infarct size or location at the acute time point. However, the same animals are being followed in a parallel study with an extended observation period, after which histological analyses will be performed to validate and complement the imaging findings.

## 5. Conclusions

There are differences in functional parameters between young and old mice in EDV, SV, and CO. Changes in strain parameters in the acute phase post-myocardial infarction do not differ significantly between young and old mice, while there is a clear deterioration in strain parameters after infarction in both groups.

## Figures and Tables

**Figure 1 diagnostics-15-01447-f001:**
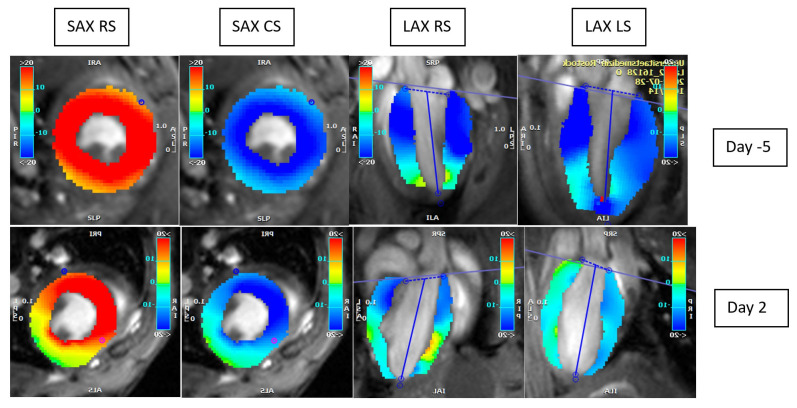
This figure shows the evaluation of the strain in an example mouse. The color visualizes the corresponding strain. Day -5 before ligature is in the upper row, and day 2 after ligature is in the lower row; SAX, short axis; LAX, long axis; RS, radial strain; LS, longitudinal strain; CS, circumferential strain.

**Figure 2 diagnostics-15-01447-f002:**
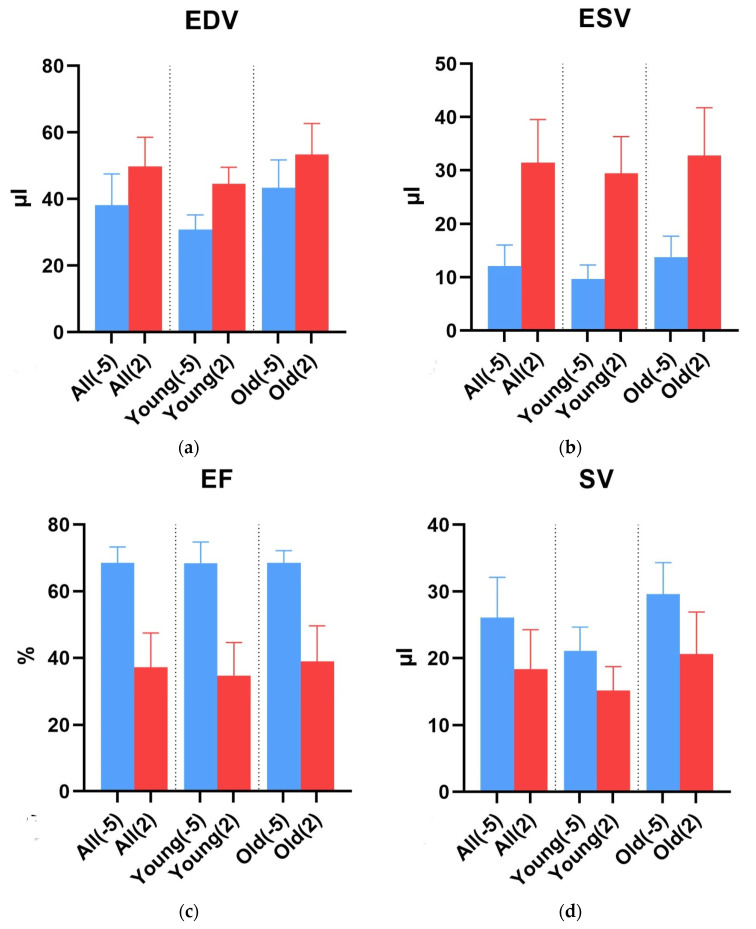
This figure shows a graphical representation in the form of bar charts of the mean and deviation values from Table 1. Day -5 before LAD ligature is blue, and day 2 after ligature is red. All charts show, for both groups combined and both groups separately, (**a**) the EDV; (**b**) the ESV; (**c**) the EF; and (**d**) the SV. EDV, end-diastolic volume; ESV, end-systolic volume; SV, stroke volume; EF, ejection fraction.

**Figure 3 diagnostics-15-01447-f003:**
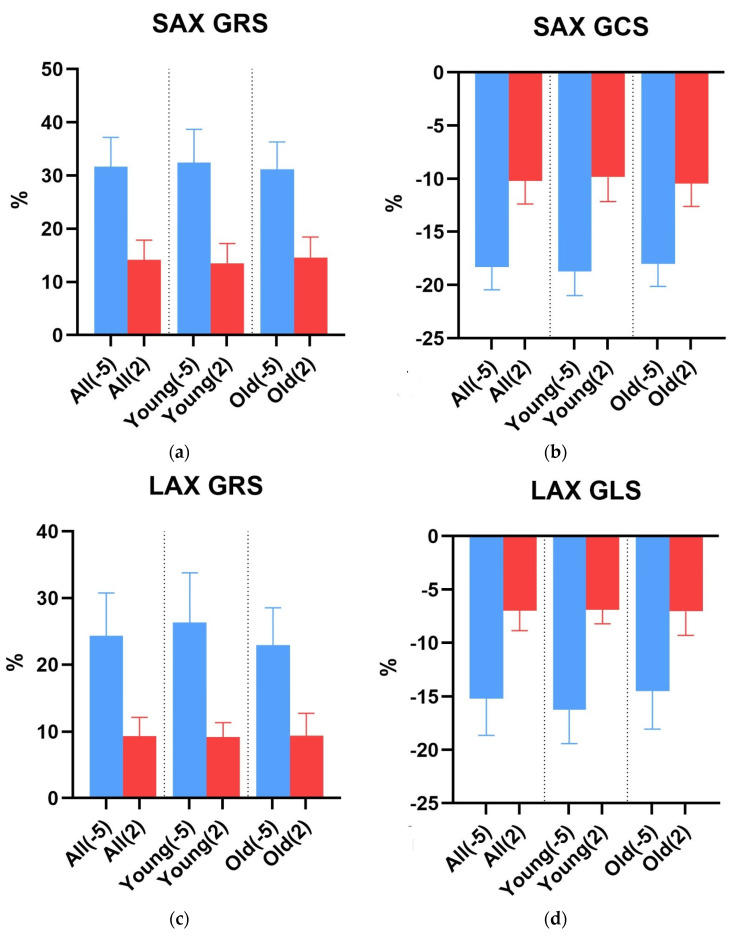
This figure shows a graphical representation in the form of bar charts of the mean and deviation values from Table 1. Day -5 before LAD ligature is blue, and day 2 after ligature is red. All charts show, for both groups combined and both groups separately, (**a**) the SAX GRS; (**b**) the SAX GCS; (**c**) the LAX GRS; (**d**) the LAX GLS. SAX, short axis; LAX, long axis; GRS, global radial strain; GLS, global longitudinal strain; GCS, global circumferential strain.

**Figure 4 diagnostics-15-01447-f004:**
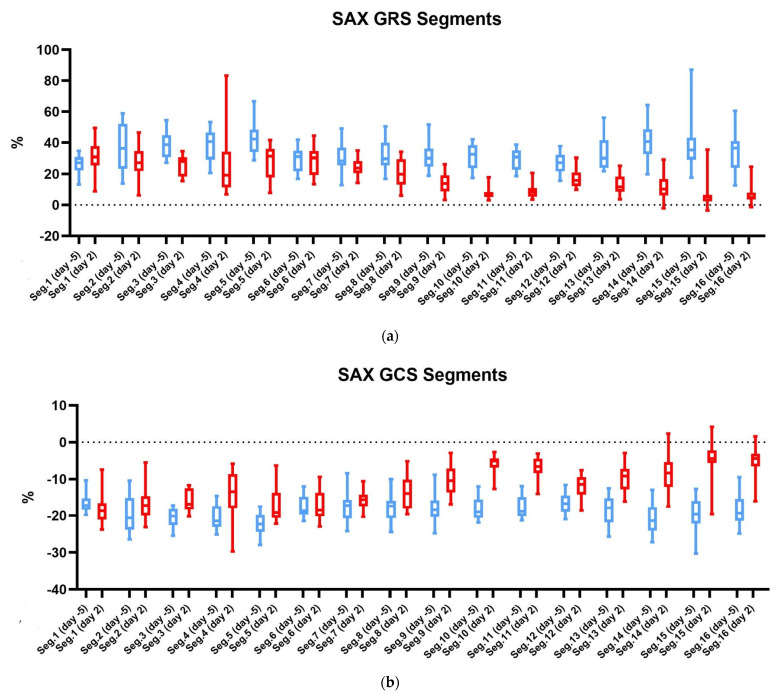
This figure shows a boxplot. Both groups of mice were combined. The strain is shown on the x-axis. The y-axis is divided into the 16 segments of the AHA model and the two time points: (**a**) the boxplot for SAX GRS; (**b**) the boxplot for SAX GCS; (**c**) the LAX GRS; (**d**) the LAX GLS. SAX, short axis; LAX, long axis; GRS, global radial strain; GLS, global longitudinal strain; GCS, global circumferential strain.

**Figure 5 diagnostics-15-01447-f005:**
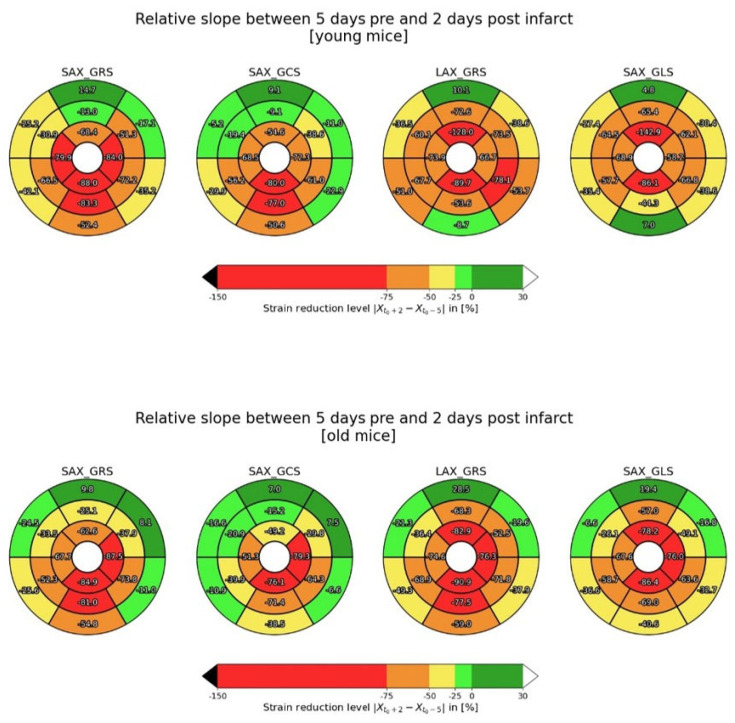
This figure shows the polar maps of young and old mice. The strain reduction level is shown for SAX GRS, SAX GCS, LAX GRS, and SAX GLS. There is one polar map for each strain for young mice in the upper row and old mice in the lower row. The percentage values are visualized as a color-coded map for each myocardial segment. SAX, short axis; LAX, long axis; GRS, global radial strain; GLS, global longitudinal strain; GCS, global circumferential strain.

**Table 1 diagnostics-15-01447-t001:** Means and Standard Deviations of Functional and Strain Parameters.

	Young		Old		All	
	MV ± SD	Delta (MV ± SD)	MV ± SD	Delta (MV ± SD)	MV ± SD	Delta (MV ± SD)
Age [days]	83.3 ± 2.9		634.8 ± 49.0		407.7 ± 282.2	
EDV day -5 [µL]	30.8 ± 4.4	13.8 ± 8.4	43.5 ± 8.3	10.0 ± 10.5	38.3 ± 9.3	11.6 ± 9.6
EDV day 2 [µL]	44.6 ± 4.9		53.4 ± 9.3		49.8 ± 8.8	
ESV day -5 [µL]	9.7 ± 2.6	19.7 ± 9.1	13.8 ± 3.9	19.0 ± 10.3	12.1 ± 3.9	19.3 ± 9.5
ESV day 2 [µL]	29.4 ± 6.9		32.8 ± 9.0		31.4 ± 8.1	
SV day -5 [µL]	21.1 ± 3.6	−5.9 ± 5.5	29.7 ± 4.7	−9.0 ± 5.4	26.1 ± 6.0	−7.7 ± 5.5
SV day 2 [µL]	15.2 ± 3.6		20.6 ± 6.3		18.4 ± 5.9	
EF day -5 [%]	68.5 ± 6.3	−33.8 ± 14.8	68.7 ± 3.6	−29.8 ± 12.4	68.6 ± 4.7	−31.4 ± 13.1
EF day 2 [%]	34.7 ± 10.0		38.9 ± 10.8		37.1 ± 10.3	
SAX GRS day -5 [%]	32.4 ± 6.3	−18.9 ± 9.7	31.2 ± 5.1	−16.6 ± 4.1	31.7 ± 5.5	−17.6 ± 6.8
SAX GRS day 2 [%]	13.5 ± 3.7		14.6 ± 3.8		14.1 ± 3.7	
SAX GCS day -5 [%]	−18.7 ± 2.2	8.9 ± 4.3	−18.0 ± 2.1	7.6 ± 1.9	−18.3 ± 2.1	8.1 ± 3.1
SAX GCS day 2 [%]	−9.8 ± 2.3		−10.4 ± 2.2		−10.2 ± 2.2	
LAX GRS day -5 [%]	26.3 ± 7.5	−17.1 ± 6.8	22.9 ± 5.6	−13.6 ± 5.7	24.3 ± 6.5	−15.0 ± 6.2
LAX GRS day 2 [%]	9.2 ± 2.2		9.4 ± 3.4		9.3 ± 2.9	
LAX GLS day -5 [%]	−16.2 ± 3.2	9.4 ± 3.0	−14.5 ± 3.6	7.4 ± 3.6	−15.2 ± 3.4	8.2 ± 3.5
LAX GRS day 2 [%]	−6.9 ± 1.3		−7.0 ± 2.2		−7.0 ± 1.9	

We listed the mean and the standard deviation, as well as the delta with the standard deviation, of all parameters separately for young and old mice and separately according to the time point. The time point day -5 is 5 days before LAD ligation, and day 2 is 2 days after LAD ligation. The heart rate did not differ significantly in relation to age or time of study. The end-diastolic volume (EDV) increased significantly in both young and old mice and was significantly higher in old mice than in the young mice at both time points. The end-systolic volume (ESV) increased significantly in both groups of mice. Before ligation, the ESV was significantly higher in old mice than in young mice. However, there was no measurable difference after ligation, although the average of 0.0328 mL was higher than that of 0.02943 mL in the young mice. The stroke volume decreased significantly after ligation in both groups of mice. A significant difference between the groups existed only before LAD ligation. The ejection fraction (EF) did not differ between young and old mice at any of the time points. In the young mice, this dropped significantly after ligation. In the old mice, it decreased but not significantly. The cardiac output (CO) was better in old mice than in young mice both before and after LAD ligation. In both groups, it decreased significantly after the procedure. The global radial strain on the short axis (SAX GRS), as well as on the long axis (LAX GLS), decreased in both groups of mice but only significantly in the young mice. The groups did not differ significantly at any time point. The global circumferential strain on the short axis (SAX GCS) also decreased in both groups of mice but only significantly in the young mice. The groups also did not differ significantly at any time point. The global longitudinal strain on the long axis (LAX GLS) decreased significantly in both groups of mice. However, again, the two groups did not differ significantly at any time point. Abbreviations: EDV, end-diastolic volume; EF, ejection fraction; ESV, end-systolic volume; GCS, global circumferential strain; GLS, global longitudinal strain; GRS, global radial strain; LAX, long axis; MV, mean value; SAX, short axis; SD, standard deviation; SV, stroke volume.

**Table 2 diagnostics-15-01447-t002:** Overview of T-Test Results and Corresponding *p*-Values.

*p*-Values	Young/Old Day -5	Young/Old Day 2	Young Day -5/Day 2	Old Day -5/Day 2
EDV	**0.002**	**0.038**	**0.005**	**0.015**
ESV	**0.029**	0.418	**0.0012**	**0.0002**
SV	**0.001**	0.06	**0.031**	**0.0005**
EF	0.951	0.475	**0.0009**	**<0.0001**
SAX GRS	0.675	0.568	**0.0021**	**<0.0001**
SAX GCS	0.508	0.595	**0.0016**	**<0.0001**
LAX GRS	0.302	0.895	**0.0005**	**<0.0001**
LAX GLS	0.308	0.873	**0.0002**	**0.0001**

The results of the *t*-tests. We tested for differences between young and old, and we tested for differences between the two days (-5 days before and 2 days after ligation) for both groups. The significant values (*p*-value < 0.05) are given in bold. Abbreviations: EDV, end-diastolic volume; EF, ejection fraction; ESV, end-systolic volume; GCS, global circumferential strain; GLS, global longitudinal strain; GRS, global radial strain; LAX, long axis; SV, stroke volume.

## Data Availability

The datasets generated and analyzed during the current study are available from the corresponding author on reasonable request.

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
