# Peer review of "Comparison of Myocardial Function in Young and Old Mice During Acute Myocardial Infarction: A Cardiac Magnetic Resonance Study"

_diagnostics, 2025, doi:10.3390/diagnostics15121447_

Round 1

Reviewer 1 Report

Comments and Suggestions for Authors

This is a well-structured and methodologically sound experimental study that investigates age-related differences in myocardial functional and strain parameters in a murine model of acute myocardial infarction (MI), using cardiac magnetic resonance imaging (CMR).

The topic is both relevant and timely, particularly given the increasing clinical focus on age-related outcomes following MI. The authors should be commended for their detailed imaging protocol, strain analysis, and transparent presentation of the results. However, certain aspects could be improved or clarified to strengthen the manuscript further.

Major Comments:

While small sample sizes are common in murine studies, the manuscript would benefit from a brief discussion of statistical power or a justification for the chosen sample size. This would help to contextualize the interpretation of non-significant findings, especially in strain parameters.

The strain parameters did not differ significantly between age groups. While the authors conclude that both young and old mice exhibit similar acute deterioration, this might be due to underpowering. A more cautious tone would be advisable here, or an acknowledgment of this limitation.

The absence of gadolinium-enhanced imaging or other measures of infarct size is a limitation. While this is acknowledged, a clearer rationale for excluding this data (e.g., focus on acute functional impact rather than tissue characterization) would improve the scientific justification.

The discussion would be strengthened by elaborating more explicitly on the potential translational implications of the findings, particularly regarding early therapeutic intervention strategies or the validity of using young animal models to simulate aged human physiology.

The visual figures (Figures 4 and 5) are compelling. However, the description in the text could more thoroughly guide the reader through the key observations. For instance, which specific segments showed the most dramatic changes and how do these correspond with expected infarct territories?

Minor Comments:

The manuscript is generally well written. A few minor grammatical improvements could enhance clarity. For example:

Line 23: “effects on functional parameters and strain differ” ;

Consider rephrasing for clarity, e.g., “to compare changes in functional and strain parameters”.

Line 36: “do not differ significantly” should be rephrased as "did not differ significantly".

Figures 2 and 3 effectively support the findings. However, consider using a more consistent color scheme and increasing font size for better readability. The explanation for abbreviations should be provided within or directly under each table for improved accessibility.

Ethical approval is appropriately mentioned. However, consider stating whether any efforts were made to minimize suffering during the experimental procedures.

Author Response

Dear Reviewer,

We would like to sincerely thank you for your careful review of our manuscript and for your constructive comments and insightful suggestions. Your feedback has been invaluable in improving the clarity and overall quality of our work.

Below, we address each of your points in detail. We have revised the manuscript accordingly and believe the changes have strengthened the study.

Major Comments:

  1. While small sample sizes are common in murine studies, the manuscript would benefit from a brief discussion of statistical power or a justification for the chosen sample size. This would help to contextualize the interpretation of non-significant findings, especially

Thank you for your thoughtful comment regarding sample size and statistical power. We would like to clarify that our study was observational in nature and aimed at providing a comprehensive, descriptive assessment of myocardial structure and function in murine models. As such, we did not designate a primary outcome measure a priori, which precluded formal power calculations typically associated with hypothesis-driven studies.

Nonetheless, we acknowledge the importance of contextualizing non-significant findings, and we have now added a brief statement to the Methods section to clarify the rationale behind our sample size and the descriptive intent of our analysis. We believe this addition will help readers better interpret the results and the scope of our conclusions.

  1. The strain parameters did not differ significantly between age groups. While the authors conclude that both young and old mice exhibit similar acute deterioration, this might be due to underpowering. A more cautious tone would be advisable here, or an acknowledgment of this limitation.

We thank the reviewer for this important observation. We agree that the lack of significant differences in strain parameters between age groups may be influenced by limited statistical power. To address this, we have revised the Discussion section to include a more cautious interpretation of the findings.

  1. The absence of gadolinium-enhanced imaging or other measures of infarct size is a limitation. While this is acknowledged, a clearer rationale for excluding this data (e.g., focus on acute functional impact rather than tissue characterization) would improve the scientific justification.

We appreciate the reviewer’s comment and agree that the absence of gadolinium-enhanced imaging or direct measures of infarct size represents a limitation of the study. We have now clarified the rationale for this decision in the revised Discussion section. Specifically, our study was designed to focus on the acute functional impact of myocardial infarction rather than on tissue characterization. Given the very early post-infarction time point examined, our aim was to assess immediate changes in myocardial function using cardiac magnetic resonance feature tracking. We have now included the explanation in the Limitations.

  1. The discussion would be strengthened by elaborating more explicitly on the potential translational implications of the findings, particularly regarding early therapeutic intervention strategies or the validity of using young animal models to simulate aged human physiology.

We thank the reviewer for this valuable suggestion. In response, we have expanded the Discussion section to more explicitly address the potential translational implications of our findings

  1. The visual figures (Figures 4 and 5) are compelling. However, the description in the text could more thoroughly guide the reader through the key observations. For instance, which specific segments showed the most dramatic changes and how do these correspond with expected infarct territories?

We thank the reviewer for the helpful suggestion. In response, we have expanded the description of Figures 4 and 5 in the Results section to guide the reader more clearly through the key observations. We now highlight the specific myocardial segments (particularly segments 9–16) that showed the most pronounced changes in strain following infarction, and we relate these findings to expected infarct territories.

Minor Comments:

  1. The manuscript is generally well written. A few minor grammatical improvements could enhance clarity. For example:
    1. Line 23: “effects on functional parameters and strain differ” ; Consider rephrasing for clarity, e.g., “to compare changes in functional and strain parameters”.

Done

  1. Line 36: “do not differ significantly” should be rephrased as "did not differ significantly".

Done

  1. Figures 2 and 3 effectively support the findings. However, consider using a more consistent color scheme and increasing font size for better readability. The explanation for abbreviations should be provided within or directly under each table for improved accessibility.

We thank the reviewer for the helpful suggestions regarding the figures and tables. In response, we have increased the figure sizes to improve readability. While we acknowledge the importance of a consistent color scheme, we are limited by the default color settings of the software used for data analysis and export. Nevertheless, we have ensured that the color coding remains clear and interpretable. Additionally, we have added explanations for all abbreviations directly below each table to enhance accessibility for the reader.

  1. Ethical approval is appropriately mentioned. However, consider stating whether any efforts were made to minimize suffering during the experimental procedures.

We thank the reviewer for raising this important point. We confirm that all efforts were made to minimize animal suffering during the experimental procedures. Mice were housed under standardized conditions with continuous access to food and water and were monitored daily by trained personnel. All procedures were performed in accordance with institutional and national guidelines for the care and use of laboratory animals. Adequate analgesia and anesthesia were administered during surgical interventions to ensure minimal discomfort. Postoperative care was provided to support recovery and ensure animal well-being throughout the study. We have clarified and specified the relevant information regarding animal welfare in the Methods section.

Reviewer 2 Report

Comments and Suggestions for Authors

Author Response

Dear Reviewer,

We would like to sincerely thank you for your careful review of our manuscript and for your constructive comments and insightful suggestions. Your feedback has been invaluable in improving the clarity and overall quality of our work.

Below, we address each of your points in detail. We have revised the manuscript accordingly and believe the changes have strengthened the study.

  1. Clarify validation of CMR strain analysis in mice with the chosen software (cite murine-specific benchmarks if available).

We thank the reviewer for this important point. In our study, we used Circle CVI for strain analysis. Although this software is widely applied in clinical and preclinical cardiac MRI, it has not been formally validated for use in murine models. However, in close consultation with the manufacturer, we adapted the volumetric and anatomical settings to account for murine cardiac dimensions and heart rates. We have now clarified this point in the Methods section and included a corresponding note in the Limitations.

  1. Feature-tracking strain has been validated primarily in humans. It is unclear whether the parameters (frame rate, spatial resolution) allow for reliable strain calculations in mice. What are the limitations of the software when working with high heart rates (~500 bpm)? Is there internal validation?

We thank the reviewer for this important comment. It is true that feature-tracking strain analysis has been primarily validated in humans, and its application in small animal models, particularly mice with high heart rates (~500 bpm), involves technical limitations. In our study, cine MRI was acquired with a spatial resolution adapted to murine cardiac anatomy. While this resolution is at the upper limit for reliable strain analysis at such high heart rates, it has been shown in previous studies to allow feasible functional assessment in mice. PMID: 34096916

As an internal consistency check, we compared the heart rate derived from the software’s analysis of cine loops with the externally measured heart rate during image acquisition. The results showed good agreement, indicating that the temporal alignment of the cardiac cycle within the cine loops was sufficiently accurate for further strain analysis. Nevertheless, we acknowledge that the lack of formal validation and absence of invasive or histological correlation represents a limitation, and we have included this in the revised Limitations section.

  1. Include representative images or overlays of strain maps pre/post-MI for visual impact.

We thank the reviewer for the helpful suggestion. Representative strain maps before and after myocardial infarction are already included in Figure 1, with the pre-ligation images (day -5) shown in the upper row and the post-ligation images (day 2) in the lower row. The corresponding strain is visualized using a color overlay. If the reviewer has a specific suggestion for additional images or alternative presentation formats, we would be happy to consider further adjustments to improve clarity and visual impact.

  1. Discuss the implications of unchanged strain between age groups—why might this differ from human studies?

We thank the reviewer for this thoughtful comment. In response, we have expanded the Discussion to address why strain values did not differ significantly between young and old mice, in contrast to observations reported in human studies. We consider possible physiological and methodological explanations and have added relevant references where applicable.

  1. Consider post-hoc power analysis to support non-significant findings.

Thank you for your thoughtful comment regarding sample size and statistical power. We would like to clarify that our study was observational in nature and aimed at providing a comprehensive, descriptive assessment of myocardial structure and function in murine models. As such, we did not designate a primary outcome measure a priori, which precluded formal power calculations typically associated with hypothesis-driven studies.

Nonetheless, we acknowledge the importance of contextualizing non-significant findings, and we have now added a brief statement to the Methods section to clarify the rationale behind our sample size and the descriptive intent of our analysis. We believe this addition will help readers better interpret the results and the scope of our conclusions.

  1. Address whether gender (all female mice) may limit generalizability or introduce bias.

We thank the reviewer for this important observation. In this study, only female mice were included in order to reduce biological variability and maintain a homogeneous experimental population. We acknowledge that this may limit the generalizability of our findings, particularly with respect to potential sex-specific differences in myocardial infarction response and cardiac remodeling. This is now acknowledged as a limitation in the revised manuscript, and we highlight the need for future studies including both sexes to further explore gender-specific responses in this model.

  1. Add more discussion on translational relevance to human aging and MI response.

We thank the reviewer for this valuable suggestion. In response, we have expanded the Discussion section to more thoroughly address the translational relevance of our findings in the context of human aging and myocardial infarction.

  1. The article only examines 1 day after LAD ligation, which reflects the acute but not the subacute or chronic stage. Explain why only this one time point was chosen and whether this limits the interpretation in terms of repair or remodeling.

We thank the reviewer for this important point. The current study was designed to focus specifically on the acute functional changes occurring shortly after myocardial infarction, as assessed by cardiac MRI one day after LAD ligation. This early time point was chosen to capture the immediate effects of ischemic injury before significant repair or remodeling processes begin. We acknowledge that this limits the interpretation with respect to subacute and chronic phases of post-infarction remodeling. However, a follow-up study using the same animals with extended observation periods and planned histological analysis is currently underway and will address these aspects in more detail.

  1. Terms such as “global circumferential strain” and “strain rate” are used but are not defined mathematically.

We thank the reviewer for this helpful comment. In response, we have expanded the Methods section to include descriptive definitions of the terms "global circumferential strain" and "strain rate" as used in the manuscript.

  1. Add a brief definition or explanation of the strain parameters used in the “Materials and Methods” section.

We thank the reviewer for the suggestion. In response, we have added brief to the “Materials and Methods” section to clarify the parameters used in our analysis.

  1. Results use statements such as “tended to be lower/higher” even though p > 0.05. Avoid drawing conclusions from insignificant differences or clearly label them as “non-significant trend.”

We thank the reviewer for this important observation. In response, we have revised the wording in the Results section to avoid implying significance where none was demonstrated.

  1. Has correction been made for multiple comparisons?

We thank the reviewer for this thoughtful comment. As the study was exploratory in nature, the primary aim was to identify potential patterns and generate hypotheses rather than to test specific predefined hypotheses. Moreover, the number of variables included in the analysis was relatively limited. For these reasons, and in line with recommendations for exploratory research, we did not apply formal corrections for multiple comparisons, which can increase the risk of Type II errors in such contexts. Nevertheless, we have interpreted the findings with appropriate caution and acknowledge the need for future confirmatory studies to validate these results. 

  1. Are there several strain parameters compared along the axis and time – was ANOVA or p-value correction used?

We thank the reviewer for this question. While multiple strain parameters were assessed at two time points, the comparisons were limited and exploratory in nature. Therefore, we used pairwise t-tests without ANOVA or formal correction. We acknowledge this as a limitation and have added a corresponding note to the Discussion.

  1. Can a relationship be made with biomarkers or histology? Currently, the results are only imaging – is there confirmation of the infarct area by TTC staining, Masson or other methods?

We thank the reviewer for this thoughtful comment. In the current study, our focus was on the acute functional response to myocardial infarction, assessed via cardiac magnetic resonance imaging. As such, no histological staining (e.g., TTC or Masson’s) was performed at this early time point. However, the same animals are being followed in an ongoing study with a longer observation period, at the end of which histological analysis will be conducted to confirm infarct localization and tissue remodeling. We have added a corresponding note to the Limitations to acknowledge this limitation and to indicate that histological validation is part of our extended study design.

  1. The reviewer asks the authors to provide better quality images of the figures. In Figure 5, in addition to the images, the captions are not clear enough.

We thank the reviewer for the helpful suggestions regarding the figures and tables. In response, we have increased the figure sizes to improve readability. Additionally, we have added explanations for all abbreviations directly below each table to enhance accessibility for the reader.